# Optimization of the Geraniol Transformation Process in the Presence of Natural Mineral Diatomite as a Catalyst

Anna Fajdek-Bieda 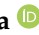

Faculty of Technology, Jacob of Paradies University, Chopina 52, 66-400 Gorzow Wielkopolski, Poland; abieda@ajp.edu.pl

**Abstract:** Process optimization is increasingly finding applications in chemical engineering. The reason for this increase in applications is to create more efficient and sustainable technological processes. Thanks to innovative models, it is possible to plan an experiment in a given field of study without much complication and carry out the optimization of such a process, achieving goals in a much shorter time period. This paper describes the performance of optimization of the geraniol transformation process in the presence of a catalyst of natural origin—diatomite. Response surface methodology (RSM) was chosen as the method. For this purpose, the following parameters were used as variables: temperature (80, 110, and 150 °C), catalyst concentration (1 wt%, 5 wt%, and 10 wt%), and reaction time (0.25 h, 12 h, and 24 h). At the same time, the functions describing the process and response functions were the conversion of geraniol (GA) as well as the selectivity of conversion to beta-pinene (BP), respectively. The obtained results made it possible to identify the optimal set of parameters at which the highest values of GA conversion and the selectivity of conversion to BP are obtained. It turned out that the GA transformation process is best carried out at 80 °C at a diatomite concentration of 1.0 wt% and a reaction time of 0.25 h.

**Keywords:** response surface methodology; geraniol; reaction time; diatomite; beta-pinene; optimization

## 1. Introduction

On the one hand, production processes are becoming more complex, and on the other hand, customers are demanding higher product quality. This situation contributes to the growing importance of optimization processes. As technologies, products and even more so requirements are changing faster and faster, engineers have much less time to learn about processes and optimize them based on their own experience. On the other hand, processes are very often automated and measured in detail so that we have a large amount of data to accurately describe processes. It turns out that data analyses can be used in process optimization in a variety of ways and can address many aspects of process management [1].

One method to significantly shorten the research cycle and again reduce the time and labor intensity of experiments is the use of mathematical experiment planning methods. Methods based on planning experiments and the mathematical processing of their results allow, especially in the absence of a broad theoretical basis, i.e., knowledge of the mechanism, kinetics, and structure, the ability to quickly create a mathematical model of the process, find the optimal development and course of the process, or select the most important parameters affecting its course [1].

In the work presented here, the response surface methodology (RSM) method was used to address the optimization problem, which consists of searching for the best possible answer as a result of producing a series of approximations in consecutive optimization steps. Response surface analysis is the step of searching for a mathematical model using the results of an experiment. It allows the mathematical model to be adjusted to the results of the experiment and the experimental plan to be changed according to the nature of the response model. The drawback of this method is the necessity of having partial information

with respect to the nature of the responses, for example, whether it is linear, quadratic, or determines other mathematical properties. The results of the design process will depend on the accuracy of the mathematical model that describes the response surface [2–4].

In practice, the response surface method is applied in a sequential manner. In the first stage, out of the total number of factors under study, it is necessary to determine those that significantly affect the value of the response surface [5]. This, in turn, directly leads to issues when planning the experiment. The execution of the experiment, followed by the interpretation of its results, makes it possible to assess the influence of the various factors studied on the value of the response surface. This phase of experimental research is usually called the zero phase of the response surface method, while the experiments performed during it are referred to as review experiments. Once the importance of various test factors has been established, one moves on to the next phase of research. In this phase, the experimenter's task is to determine whether the current values of independent variables (nominal values) correspond to the extremes of the response surface. If the current values of the factors under study do not correspond to the optimum of the response function, it is necessary to take additional measures to move the process toward the neighborhood of the optimum point. During this phase of process improvement, the method of greatest decline is usually used. This optimization technique makes extensive use of fractional bivariate experimental plans. The purpose of using this type of experimentation is to determine the local tendencies of changes in the response surface described using first-degree polynomials. If the process is close to an optimum, the second and final phase of the experimental optimization is performed. Its purpose is to obtain the most accurate description of the process around the point corresponding to the extreme. The region of the factor space covered by the experiment must be narrow enough to obtain an accurate approximation of the response function. Since the optimization procedure has reached the neighborhood of the optimum, a curvature effect is to be expected. Making the above assumptions, in many cases a sufficiently good approximation of the response function is a second-degree polynomial. An adequate process model then makes it possible to determine the optimal conditions for running the process due to the quality criterion adopted [6]. To date, the RSM method has been used in the waterjet treatment of rock materials [7–9] and heavy metals [10,11].

RSM is one of the more commonly used metamodeling methods, the purpose of which is to approximate the response of the model on the basis of the selected values of input signals [5]. The problem of approximating any function reduces into a search for the function, $g$, described by Equation (1):

$$\hat{y} = g(x) \tag{1}$$

which yields values that are closest to the modeled real f function. The most common approach is to narrow the search to a certain class of mappings that is general enough not to be too limiting in its ability to fit the data set. The simplest way to carry this out is to establish a certain number of functions $g1, \dots, gM$ along with the assumption that the function, $g$, describing the process being modeled can be represented as some combination of them, as shown in Equation (2).

$$g(x) = \sum_{k=1}^{M} a_k g_k(x) \tag{2}$$

That is, the $g_k$ functions are determined, while the corresponding $a_k$ coefficients are unknown. For these coefficients to be unambiguously determined, the $g_k$ functions should be linearly independent (basis functions). In the response surface method, basis functions are restricted to certain selected classes. Most often, polynomials are taken as basis functions,

while striving to use the lowest possible degree. For small curvatures of the response surface, first-order polynomials of the form in Equation (3) are used:

$$g(x) = \beta_0 + \sum_{i=1}^{n} \beta_i x_i \tag{3}$$

where $n$ denotes the size of the vector of input variables $x$. For significant curvature, the response surface is approximated by second-order polynomials of the following form Equation (4).

$$g(x) = \beta_0 + \sum_{i=1}^{n} \beta_i x_i + \sum_{i=1}^{n} \beta_{ii} x_i^2 + \sum_{i=1}^{n} \sum_{j=1; i=1}^{n} \beta_{ij} x_i x_j \tag{4}$$

The $\beta_i$ parameters of the polynomials can be determined by regression analyses, fitting the best possible approximating function to the data at hand. Another class of basis functions, often used in the response surface method, is the radial basis function (RBF) group. The most commonly used radial functions are as follows:

- The Gaussian function;

$$g_k(x) = e^{-cx^2}, 0 < c \le 1 \tag{5}$$

- The quadratic function of the inverse form;

$$g_x(x) = \sqrt{x^2 + c^2}, 0 < c \le 1 \tag{6}$$

- Quadratic function.

$$g_k(x) = \frac{1}{x^2 + c^2}, 0 < c \le 1 \tag{7}$$

One of the innovative methods of multiple response optimizations used in different industry branches is response surface methodology (RSM). It is a frequently used method thanks to its negligible number of needed tests and inexpensive research. Furthermore, in most cases, there is more than one significant response, which causes problems that must be optimized simultaneously. This is more difficult especially when existing objectives conflict with one another.

In recent years, the method has gained popularity in optimizing biochemical processes such as the enzymatic synthesis of fatty esters or the hydrolysis of pectic substrates, the synthesis of butylgalactoside by galactosidase, or the biotransformation of 2-phenylethanol to phenylacetone aldehyde [12]. One of the applications of the RSM method is its use in optimizing the production of alkaline proteases, the effect of casamino acid concentration, glucose concentration, inoculum age, incubation time, and stirring speed [13]. Senanayake and Shahidi [14] studied the biochemical reaction catalyzed by lipase. They evaluated the effects of three different independent parameters, i.e., the amount of enzymes, reaction temperature, and reaction time, on the efficiency of DHA (docosahexaenoic acid) incorporation. Reference [15] attempted to optimize the hydrolysis process with respect to the efficiency of pectolytic enzymes. Lee et al. [16] improved the cholesterol oxidase preparation process with RSM. Cholesterol concentration (g/L), yeast extract (g/L), and Tween (mL/L) were selected as individual parameters. A central complex design and a second-order polynomial style equation were developed, and a polynomial calculation was fitted to the new data. Gobbetti et al. [17] studied the effects of temperature, pH, and NaCl on peptidase activity in lactic acid bacteria using RSM. The effects of each factor were studied within the parameter ranges applicable to cheese ripening.

Several valuable compounds can be formed during the transformation of geraniol. These can be products of isomerization (linalool and nerol), dehydration (β-pinene and ocimene), oxidation (geranial and neral), and dimerization, and the products of the cyclization and fragmentation of the carbon chain are 6,11-dimethyl-2,6,10-dodecatrien-1-ol

(2E,6E)-6,11-dimethyl-2,6,10-dodecatrien-1-ol and thumbergol. This is evidenced by the publications described below on the transformation process of geraniol using various minerals of natural origins.

Reference [18] presents the modeling of the geraniol transformation process using response surface methodology (RSM). The influences of key process parameters such as temperature 20–110 °C, catalyst concentration (mironekuton) 1.0–5.0 wt%, and reaction time 0.25–2 h were studied. The response functions were the conversion of geraniol (GA), the selectivity of conversion to thumbergol, and the selectivity of the conversion to 6,11-dimethyl-2,6,10-dodecatriene-1-ol.

Reference [19] presents the results of studies undertaken for the conversion of geraniol in the presence of pomegranate as a catalyst using the response surface method (RSM). In this method, the effects of the following parameters were analyzed: temperature 50–150 °C, catalyst (pomegranate) concentration 1.0–10.0 wt%, and reaction time 0.25–24 h. Response functions included the conversion of geraniol, selectivity to neral, and selectivity to citronellol. For neral, the optimal amount of selectivity was achieved at 49 mol% at 60 °C, a catalyst concentration of 2.5 wt%, and a reaction time of almost 2 h. For citronellol selectivity, the optimal value of 49 mol% was achieved for the following control factors: temperature of 20 °C, catalyst concentration of 5.0 wt%, and reaction time of 2 h. The optimal set of control factors was 55 °C with respect to temperature, a catalyst concentration of 5.0 wt%, and a reaction time of 2 h.

The proposed transformation of geraniol in the presence of diatomite was not optimized. With the proposed method, optimal factors such as temperature (50–150 °C), the amount of catalyst (1–10 wt%), and reaction time (0.25–24 h) can be selected, taking into account high values of geraniol conversion and the selectivity of conversion to beta-pinene (Figure 1).

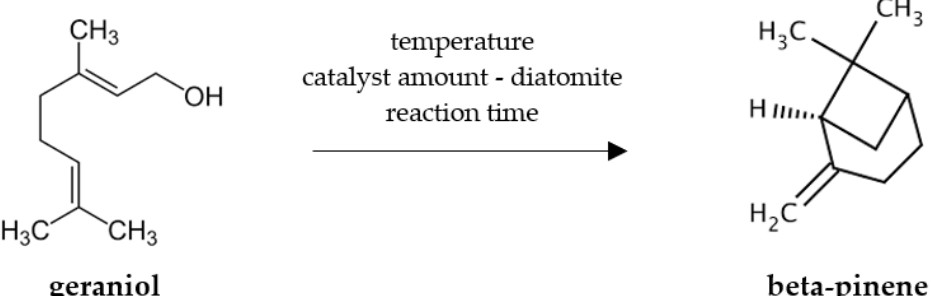

**Figure 1.** Reaction scheme for the transformation of geraniol to beta-pinene.

## 2. Results and Discussions

### 2.1. Influence of Process Parameters on GA Conversion

The morphology and microstructure of natural diatomite are shown in Figure 2. The mineral is characterized by an ordered micro- and nanoporous structure. The picture shows the centric (discoidal) clusters of diatoms. Some of the centric diatoms visible in the photo have a radius of about 20 μm. In addition, numerous clusters of diatom shells of smaller sizes are visible. In addition, the structure of the diatomite is characterized by significant porosity. Based on the scanning image, it can be concluded that the diatomite has a large pore volume in addition to a highly porous structure. The high porosity of this material was one of the main reasons for selecting it as a potential catalyst in the transformation of geraniol.

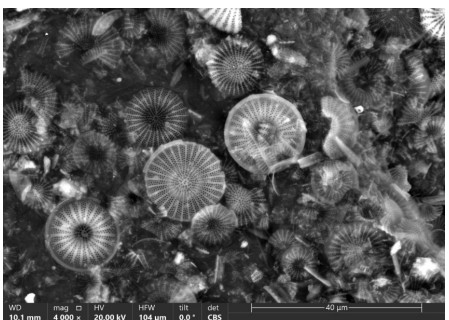

**Figure 2.** SEM (scanning electron microscopy) image of diatomite.

Figure 3 shows the results of the elemental analysis for the diatomite. The highest concentrations were obtained for carbon and oxygen, which were 54.5% and 41%, respectively. In addition, the presence of Si (4.2%) and aluminum at about 0.2% was confirmed.

**Elemental Composition of Diatomite (%)**

| C | O | Al | Si |
|---|---|----|----|
| 54.5 | 41.1 | 0.2 | 4.2 |

Colour code: #C58600

Colour code: #6E0000

Colour code: #004B7C

Colour code: #258306

**Figure 3.** Element maps for diatomite.

The FTIR spectrum obtained for the diatomite shows characteristic adsorption bands at 3436, 1625, 1094, and 797 cm$^{-1}$ (Figure 4). The band at 3436 cm$^{-1}$ confirms the presence of the free silanol group (SiO-H), 1625 cm$^{-1}$ characterizes the bending vibration of H-O-H, 1094 cm$^{-1}$ corresponds to the stretching vibration of the siloxane group (-Si-O-Si), and 797 cm$^{-1}$ is characteristic for the vibration of the SiO-H group.

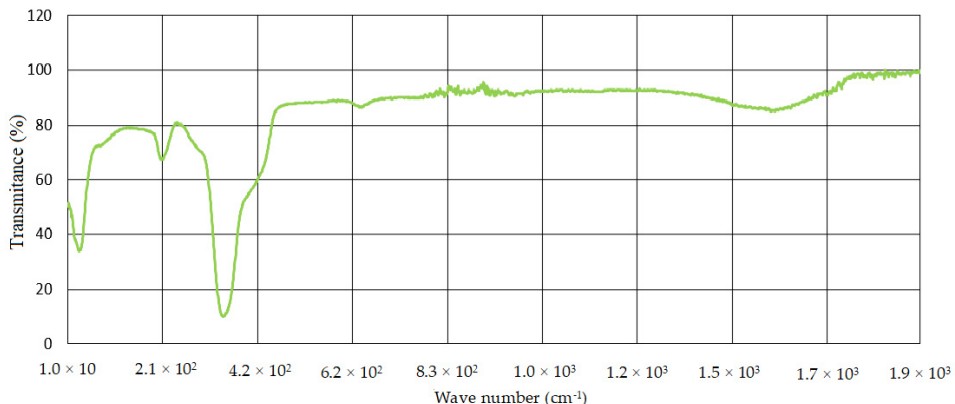

**Figure 4.** FT-IR (infrared spectroscopy) spectra of diatomite.

Figure 5 shows the XRD spectrum of natural diatomite. The obtained diffractogram indicates that the diatomite consists mainly of silica (SiO$_2$) with small amounts of Al$_2$O$_3$, Fe$_2$O$_3$, CaO, and Na$_2$O. The most abundant phases in the sample were minerals from the group of quartz, muscovite, mica, and clay minerals (mainly kaolin). The presence of an amorphous phase was significant within the area from 4 to 20° 2θ and is probably the result of SiO$_2$ glass formation. Characteristic SiO$_2$ peaks appeared at 19°, 21°, 26°, and 35°. The XRD examination showed that the diatomite was poorly crystallized.

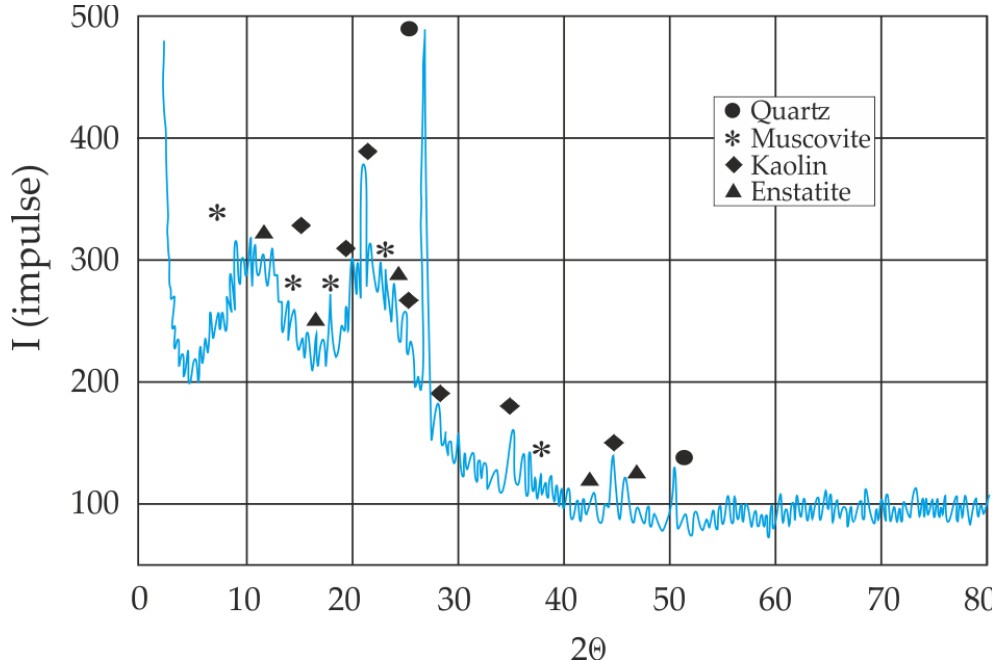

**Figure 5.** X-ray (X-ray diffraction) diffraction data for diatomite.

In order to obtain a complete morphological description of the diatomite mineral used, its BET surface area and total pore volume were analyzed. The analyzed mineral is characterized by a larger surface area of 33.6 m$^2$/g and a larger pore volume of 0.12 cm$^3$/g relative to the literature's data.

## 2.2. Influence of Process Parameters on GA Conversion

The correct analysis is presented for a 95% confidence level ($\alpha = 0.05$). It turns out that the simulation coefficient is significant when it reaches a $p$-value $< 0.05$. In addition, the correlation coefficient ($R^2$) and the corrected correlation coefficient ($R^2$adj) were determined to verify the precision of the model. According to the analysis, the $R^2$ figure was 0.971, and $R^2$adj was 0.956. Accordingly, the model described 92.33% of the variability in the data. In addition, the variation between $R^2$ and $R^2$adj was less than 0.2 for all response variables, suggesting that the response surface correctly describes the data (Table 1).

**Table 1.** Analysis of variance of the GA conversion.

| Source | DF | Adj SS | Adj MS | F-Value | *p*-Value | VIF |
|---|---|---|---|---|---|---|
| Model | 9 | 17,227.9 | 1914.21 | 64.40 | 0.000 | |
| **Linear** | 3 | 13,500.5 | 4500.16 | 151.41 | 0.000 | 1.01 |
| Temperature (°C) | 1 | 6633.5 | 6633.47 | 223.19 | 0.000 | 1.01 |
| Catalyst concentration (wt%) | 1 | 5794.3 | 5794.35 | 194.95 | 0.000 | 1.01 |
| Time (h) | 1 | 1107.3 | 1107.27 | 37.25 | 0.000 | 1.01 |
| **Square** | 3 | 3557.6 | 1185.85 | 39.90 | 0.000 | 1.00 |
| Temperature (°C) × Temperature (°C) | 1 | 3423.6 | 3423.57 | 115.19 | 0.000 | 1.00 |
| Catalyst concentration (wt%) × Catalyst concentration (wt%) | 1 | 106.5 | 106.52 | 3.58 | 0.075 | 1.01 |
| Time (h) × Time (h) | 1 | 27.5 | 27.46 | 0.92 | 0.350 | 1.00 |
| **2-Way Interaction** | 3 | 1433.5 | 477.82 | 16.08 | 0.000 | 1.00 |
| Temperature (°C) × Catalyst concentration (wt%) | 1 | 1251.6 | 1251.65 | 42.11 | 0.000 | - |
| Temperature (°C) × Time (h) | 1 | 50.8 | 50.79 | 1.71 | 0.209 | 1.01 |
| Catalyst concentration (wt%) × Time (h) | 1 | 131.0 | 131.03 | 4.41 | 0.051 | 1.01 |
| Error | 17 | 505.3 | 29.72 | - | - | 1.01 |
| Total | 26 | 17,733.2 | - | - | - | 1.01 |

GA—geraniol; DF = degree of Freedom; Adj SS = adjusted sums of squares; Adj MS = adjusted mean squares; F value is a value on the F distribution; $p$-value—$p$-value or test probability; VIF—variance inflation factor.

The standardized effects of all analyzed individual variable quantities and the relationship between them are shown in Figure 6. The standardized effect is the minimum size for displaying the effect of each variable quantity. The range of the effect refers to the size of a given quantity. The individual quantity variables have a statistically significant effect and play a significant role in the response if the bar of the normalized score exceeds the minimum ceiling, which in this case is 2.11, and it is shown as a perpendicular red line.

In order to approximate multicollinearity, the variance increase factor (VIF) was determined. Based on it, the strength of multicollinearity is calculated. VIF reveals how much the variance of the evaluated regression factor is excessively increased due to the presence of multicollinearity in the model. When the VIF is 1.0, multicollinearity is not present. For all analyzed factors, no significant multicollinearity was observed, as the VIF belongs to interval {1, 1.01}.

In order to check the adequacy of the model, so-called residual diagrams were made (Figure 7). As observed from the diagrams shown, the response model is constant. This means that neither the change in responses had an impact nor was there any apparent effect on normalization. As observed in the "Versus Fits" graph, there is no fixed pattern suggesting that the variance of the original interpretations is constant for all response values. Similarly, the histogram of residuals shows that residuals have a normal effect for all response values. In summary, all graphs in Figure 4 showed that the model is appropriate for the geraniol transformation process.

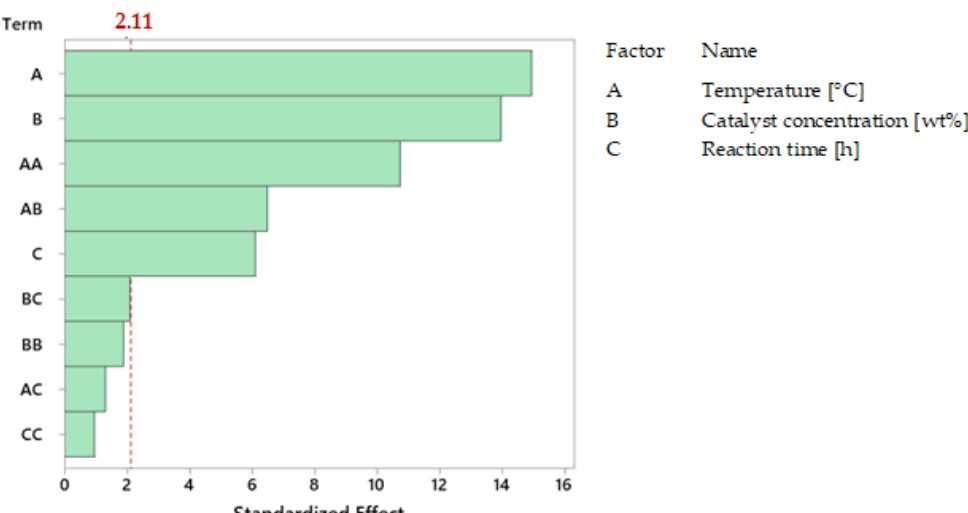

**Figure 6.** Pareto chart of standardized effects; response is GA (geraniol) conversion (mol%) ($\alpha = 0.05$).

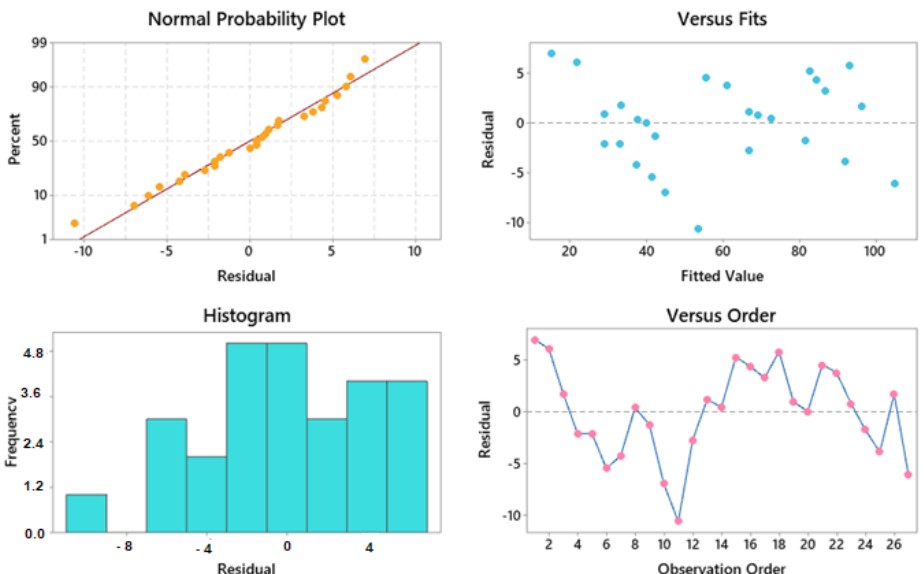

**Figure 7.** Residual plots for conversion of GA (geraniol) (mol%).

Founded on these sound effects, the subsequent computation of GA conversion was necessary:

$$C_{GA} = -240.1 + 4.728T - 0.35C_c + 0.066t - 0.019970T^2 - 0.211C_c^2 - 0.0152t^2 - 0.06449TC_c - 0.00439Tt - 0.0617C_ct \quad (8)$$

where

$C_{GA}$    denotes geraniol conversion (mol%);
$T$    denotes temperature (°C);
$C_c$    denotes catalyst concentration (wt%);
$t$    denotes reaction time (min).

Figure 8 shows the technological parameters with respect to the GA conversion function. Both an increase in temperature and catalyst concentration causes a significant increase in GA conversion values. For temperatures that are already within the range of about 110 °C to 150 °C, conversion values reach a maximum of >80 mol%. A similar relationship can be observed for the catalyst concentration, for which the GA conversion function assumes a maximum of >90 mol% within the range from about 6 wt% to 10 wt%. For

reaction time, the GA conversion function takes on low values for short time periods. As reaction time increases, GA conversion values increase to about >70 mol%.

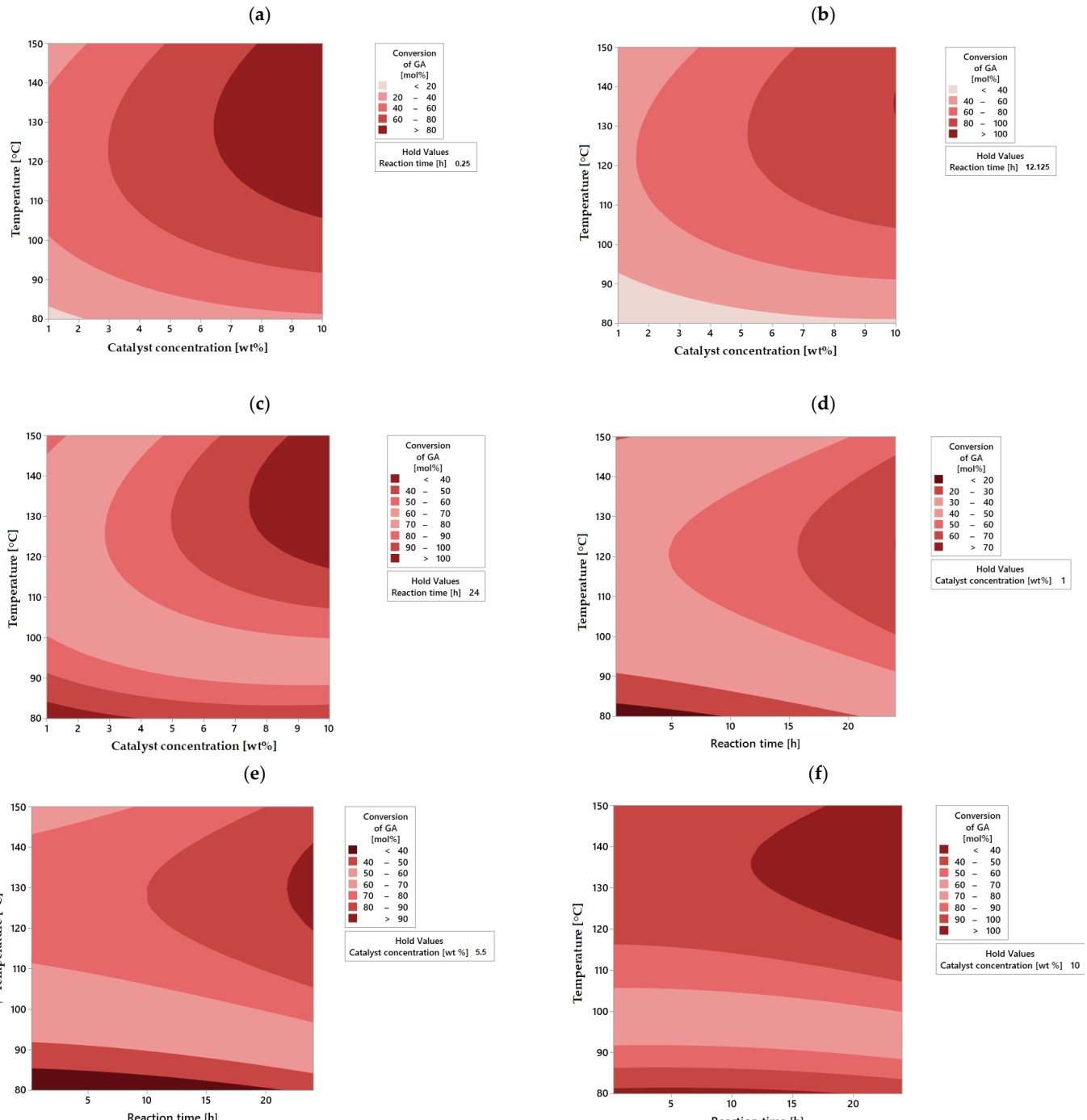

**Figure 8.** *Cont.*

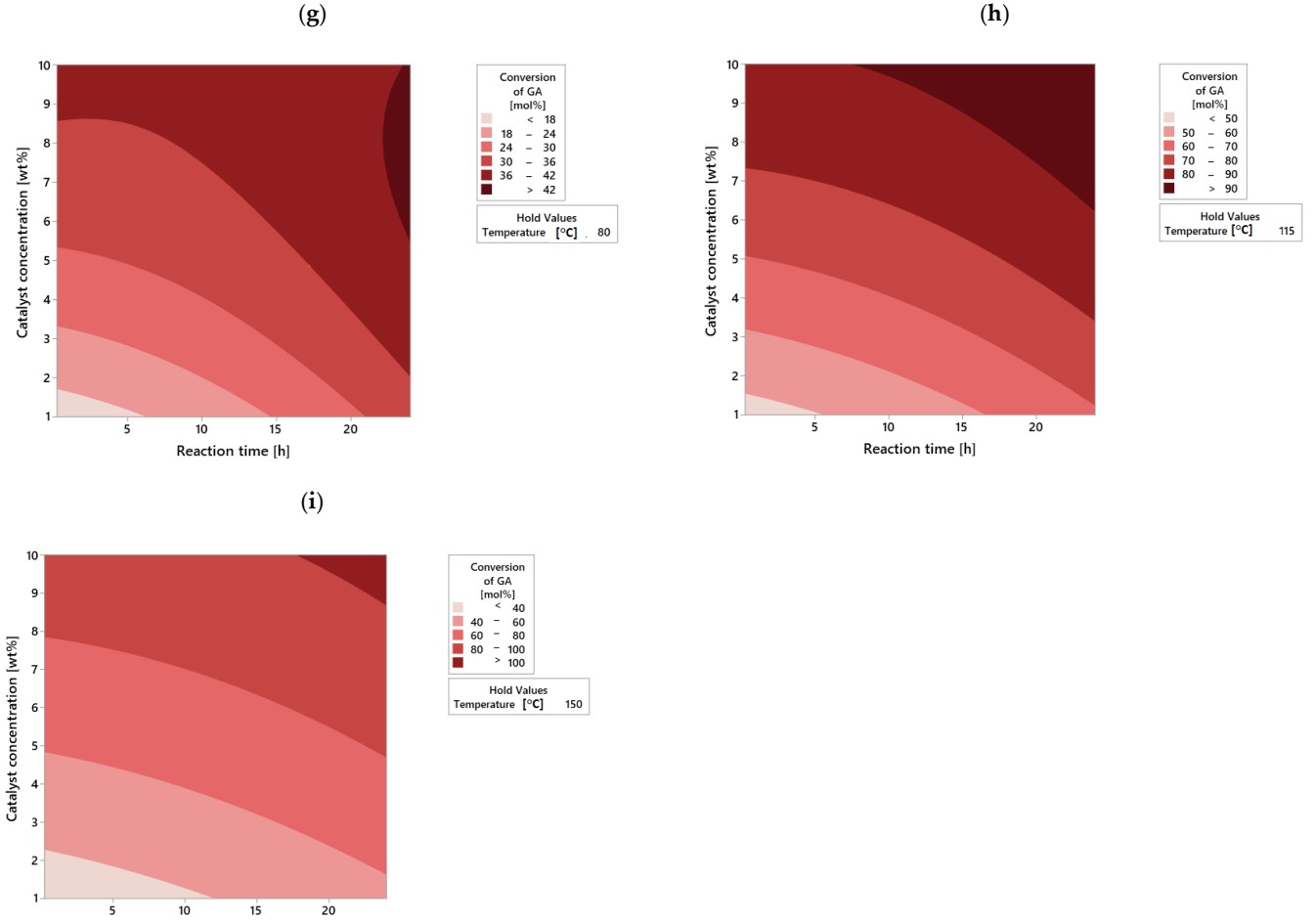

**Figure 8.** The influence of transformation process parameters on geraniol conversion ($C_{GA}$) at: time: (**a**) 0.25 h, (**b**) 12.125 h, and (**c**) 24 h, concentration: (**d**) 1 wt%, (**e**) 5.5 wt%, and (**f**) 10 wt%, and temperature: (**g**) 80 °C, (**h**) 115 °C and (**i**) 150 °C.

### 2.3. Influence of Process Parameters on β-Pinene (BP) Selectivity

A comprehensive BP selectivity study was achieved by ANOVA for 95% confidence at $\alpha = 0.05$ (Table 2). The components of the brand were deemed important when their *p*-values exceeded 0.05. Here, the $R^2$ parallel measurement and $R^2$adj varied parallel measurement were used to determine the precision of the model. The $R^2$ factor was 0.977, and $R^2$adj was 0.965, as demonstrated in Table 2. Therefore, the prototype described 93.57% of the variation of the data.

The standardized effect of all individual quantitative variables analyzed and the relationships between them are shown in Figure 9. It turns out that the individual quantitative variables have a statistically significant effect and play a significant role in the response if the bar of the standardized effect exceeds the minimum ceiling, which in this case is 2.11, and it is shown as a perpendicular red line.

In order to approximate multicollinearity, the variance increase factor (VIF) was determined. It calculates the strength of the multicollinearity phenomenon. VIF reveals how much the variance of the evaluated regression factor is inflated due to multicollinearity in the model. When the VIF is 1.0, multicollinearity is not present. For all analyzed factors, no significant multicollinearity was observed, as the VIF belongs to interval {1, 1.01}.

**Table 2.** Analysis of Variance of BP selectivity.

| Source | DF | Adj SS | Adj MS | F-Value | *p*-Value | VIF |
|---|---|---|---|---|---|---|
| Model | 9 | 10240.2 | 1137.81 | 82.75 | 0.000 | 1.01 |
| **Linear** | 3 | 8423.6 | 2807.87 | 204.21 | 0.000 | 1.01 |
| Temperature (°C) | 1 | 4324.0 | 4323.97 | 314.47 | 0.000 | 1.01 |
| Catalyst concentration (wt%) | 1 | 2556.5 | 2556.47 | 185.92 | 0.000 | 1.01 |
| Time (h) | 1 | 1562.5 | 1562.54 | 113.64 | 0.000 | 1.00 |
| **Square** | 3 | 2294.5 | 764.85 | 55.62 | 0.000 | 1.00 |
| Temperature (°C) × Temperature (°C) | 1 | 2158.6 | 2158.56 | 156.98 | 0.000 | 1.01 |
| Catalyst concentration (wt%) × Catalyst concentration (wt%) | 1 | 128.8 | 128.76 | 9.36 | 0.007 | 1.00 |
| Time (h) × Time (h) | 1 | 7.2 | 7.22 | 0.53 | 0.479 | 1.00 |
| **Two-Way Interaction** | 3 | 176.9 | 58.97 | 4.29 | 0.020 | 1.01 |
| Temperature (°C) × Catalyst concentration (wt%) | 1 | 129.0 | 128.98 | 9.38 | 0.007 | 1.01 |
| Temperature (°C) × Time (h) | 1 | 27.6 | 27.55 | 2.00 | 0.175 | 1.01 |
| Catalyst concentration (wt%) × Time (h) | 1 | 20.4 | 20.38 | 1.48 | 0.240 | 1.01 |
| Error | 17 | 233.8 | 13.75 | | | |
| Total | 26 | 10474.0 | | | | |

BP—β-pinene; DF = degree of Freedom; Adj SS = adjusted sums of squares; Adj MS = adjusted mean squares; F value is a value on the F distribution; *p*-value—*p*-value or test probability; VIF—variance inflation factor.

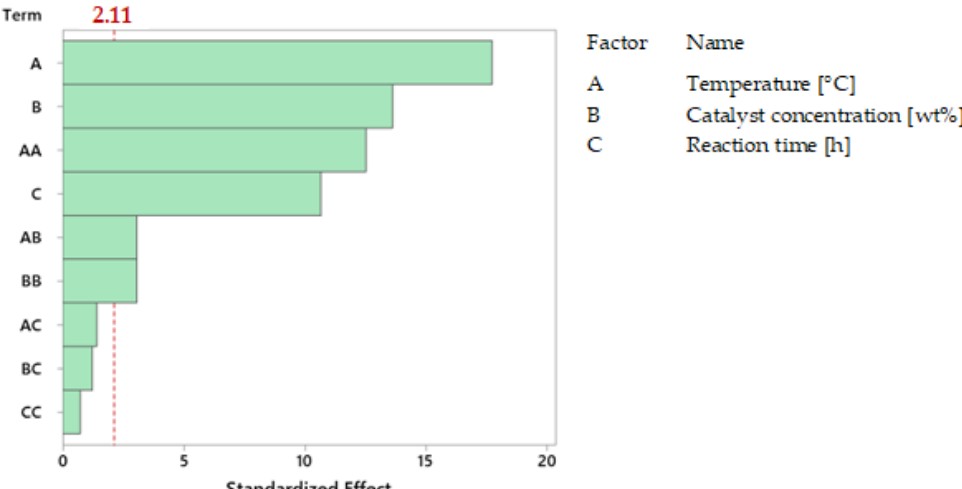

**Figure 9.** Pareto chart of standardized effects; response is BP selectivity (mol%) ($\alpha$ = 0.05).

Residual schemes were made to check the adequacy of the model (Figure 10). As observed in the diagrams given, the reply model was fixed for a normal supply. This means that neither a response change was needed nor was there any apparent normalcy problem. As observed in the "Versus Fits" graph, there is no regular pattern suggesting that the variance of the original interpretations is constant for all response values. Similarly, the histogram of the residuals shows that the residuals have a normal supply for all observations. In summary, all graphs in Figure 6 show that the model is appropriate for the geraniol transformation process.

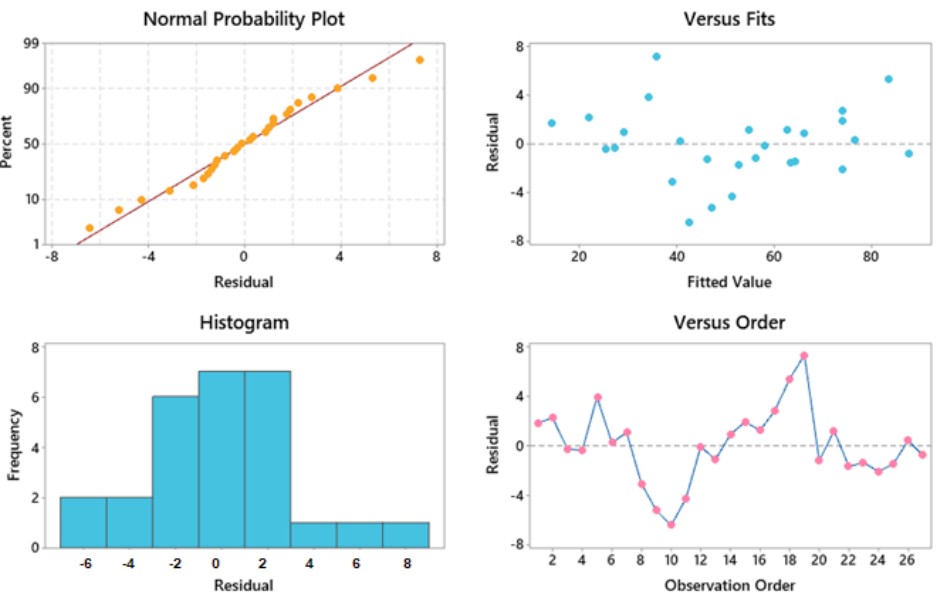

**Figure 10.** Residual plots for the selectivity of β-pinene (mol%).

On the basis of the studied causes, the subsequent calculation is described:

$$C_{BP} = -203.1 + 3.933T + 2.53C_c + 0.424t - 0.01586T^2 - 0.2321C_c^2 - 0.0078t^2 - 0.02070TC_c - 0.00363Tt - 0.0243C_ct \quad (9)$$

where

$C_{BP}$  denotes BP selectivity (mol%);
$T$  denotes temperature (°C);
$C_c$  denotes catalyst concentration (wt%);
$t$  denotes reaction time (min).

Figure 11 shows the effect of technological parameters on the course of BP selectivity values. For each parameter, the same trend can be observed in that the function values increase as the parameter increases. In the case of temperature, an increase in BP selectivity values can be observed starting at around 100 °C. Similarly, in the case of catalyst concentration, the selectivity function reaches a maximum within the range from 9 to 10 wt%. Increasing the reaction time to about 20 h causes the BP selectivity function to reach maximum values >60 mol%.

(**a**)                                                                    (**b**)

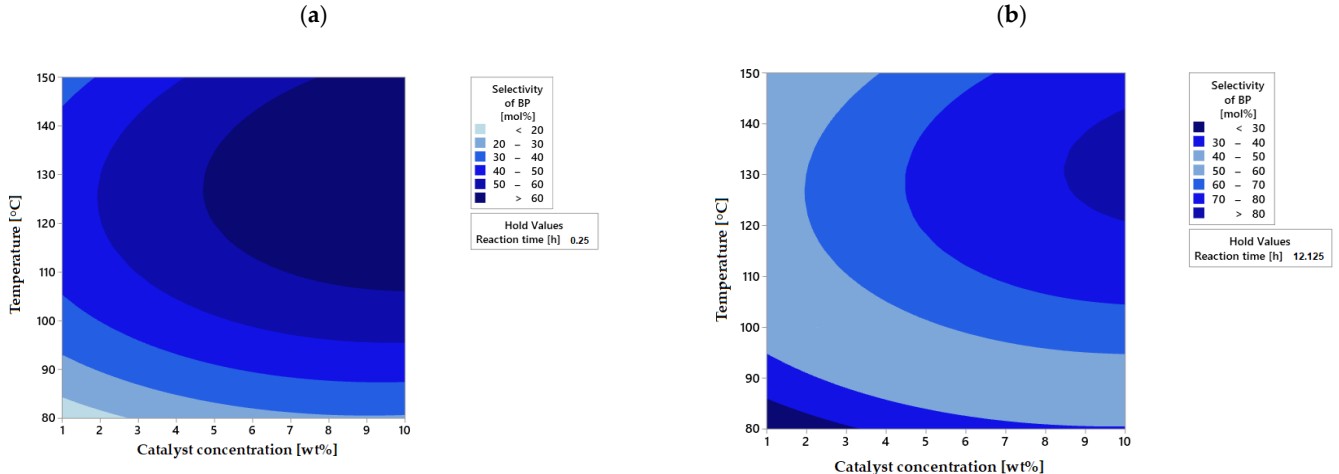

**Figure 11.** *Cont*.

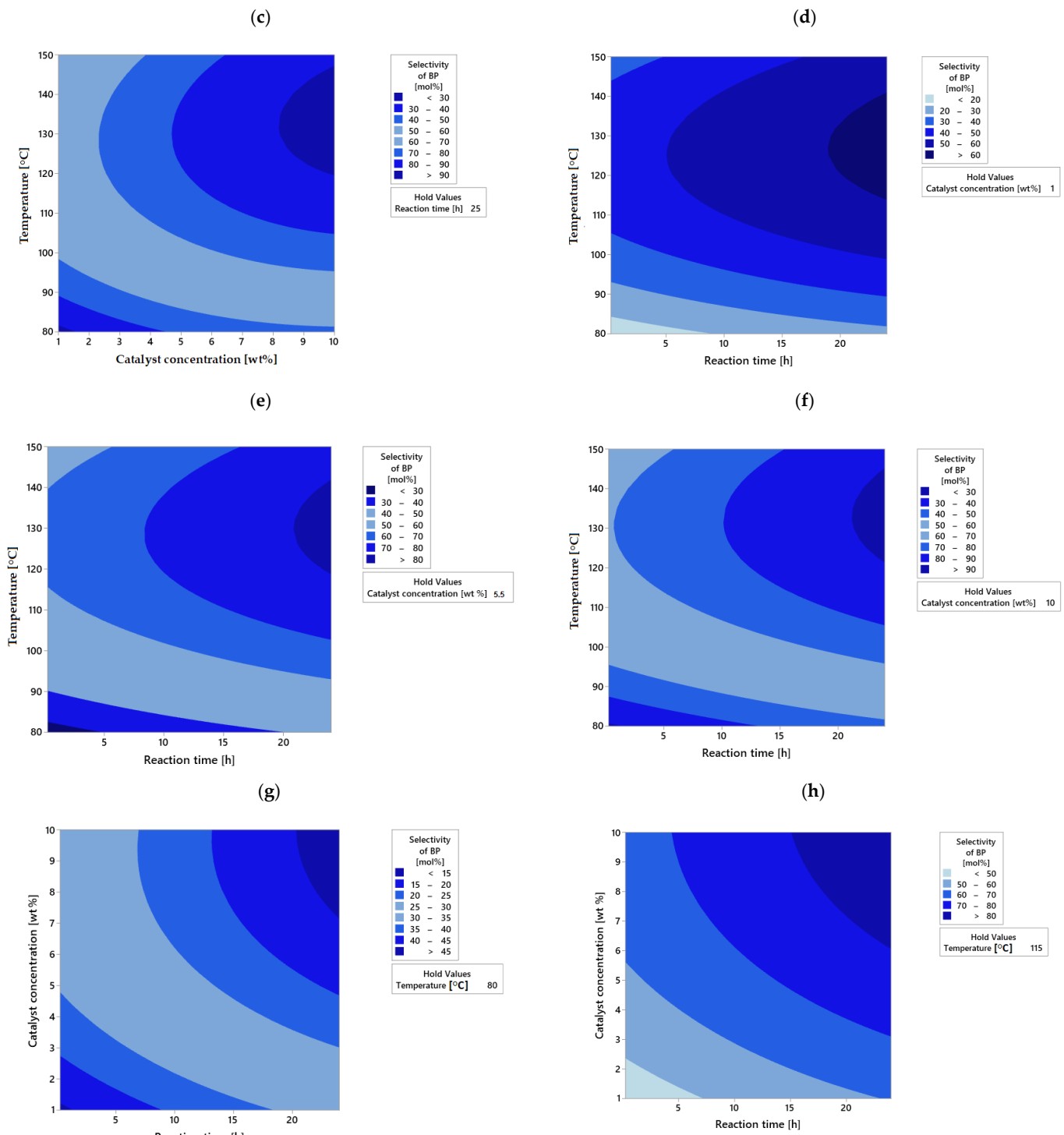

**Figure 11.** *Cont*.

**(i)**

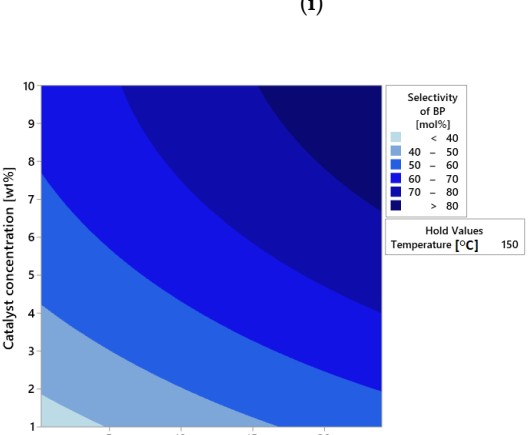

**Figure 11.** The influence of transformation process parameters on β-pinene selectivity at: time: (**a**) 0.25 h, (**b**) 12.125 h, and (**c**) 25 h, concentration: (**d**) 1 wt%, (**e**) 5.5 wt%, and (**f**) 10 wt%, and temperature: (**g**) 80 °C, (**h**) 115 °C and (**i**) 150 °C.

The use of higher temperatures, higher catalyst contents, and longer reaction times would result in the formation of other products that are less desirable, which would only constitute impurities (oxidation products and polymer compounds) and definitely affect the quality of the process itself. At higher temperatures, mainly oligomeric compounds are formed (a sign of this may be the increased darkening of the post-reaction mixture as the time of the transformation process increases).

*2.4. Composite Desirability*

The results of individual control parameters on all output parameters computed on ground Equations (8) and (9) are shown in Figure 12. In addition, individual (for each output factor) and composite desirability were assessed. Individual and composite desirability estimates how well a variable fulfils the defined reaction targets.

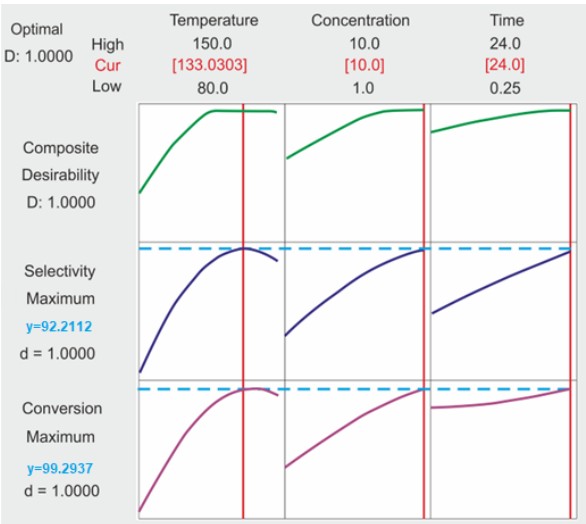

**Figure 12.** Combination of control parameter values at the optimal condition for all output parameters (cur—current).

Individual desirability (d) shows how the sets optimize a single response; composite desirability (D) appraises how the settings optimize a set of responses overall. Desirability has a range from 0 to 1. A composite desirability close to 1 indicates that settings appear to achieve favorable results for all responses as a whole.

In this case, the composite desirability is equal to 1 (or almost 1), which implies that the settings appear to reach reasonable results for all responses. On this chart bases, the optimums for all output parameters were specified. The best set of control parameters (temperature of 133 °C, catalyst concentration of 10.0 wt%, and time of 24 h) is indicated by vertical red lines.

In order to finally verify the correctness of the adopted regression models and the optimal parameters of the studied process determined on their basis, an experiment was performed under the optimal conditions obtained according to the RSM method. For the experiment, the values of the main functions describing the process, that is, the conversion of geraniol and the selectivity of conversion to β-pinene, were determined. Then, the experimental values were compared with the predicted ones (Figure 13). It was found that the obtained results in most cases fell within the limits set by the corresponding confidence intervals, which confirmed the correctness of the adopted regression models and the choice of optimal process parameters.

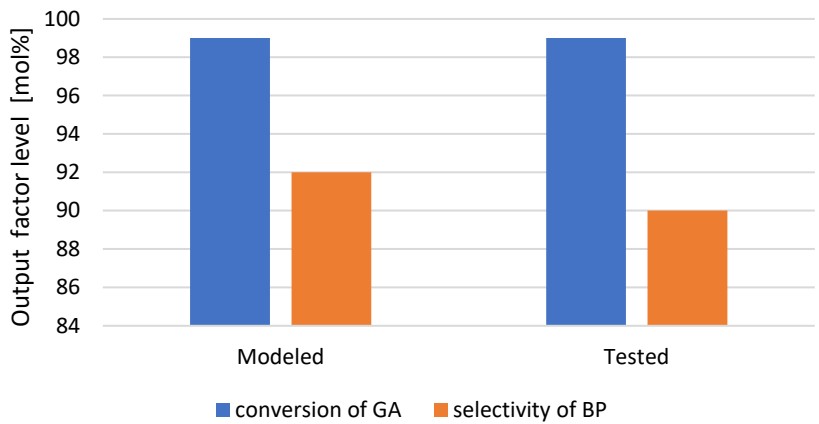

**Figure 13.** Comparison of modelled and tested output parameters of optimization.

## 3. Materials and Method

### 3.1. Raw Materials

Syntheses were carried out in the presence of diatomite (100% pure, Nanga, Hobbs, NM, USA) as a catalyst. The organic raw material used in the study was GA (99% pure, from Acros Organics, Milwaukee, WI, USA). For quantitative analysis, which was carried out by gas chromatography (GC), the standards were used as follows: citronellol (95% pure, from Sigma Aldrich, Steinheim, Germany), citral (95% pure, from Sigma Aldrich, Steinheim, Germany), ocimene (90% pure, from Sigma Aldrich, Milwaukee, WI, USA), beta-pinene (95% pure, from Fluka, Milwaukee, WI, USA), linalool (97% pure, from Acros, Steinheim, Germany), farnesol (96% pure, from Acros, Steinheim, Germany), nerol (97% pure, from Acros, Steinheim, Germany), myrcene (technically pure, from Sigma Aldrich, Steinheim, Germany), and geranylgeraniol (85% pure, from Sigma Aldrich, Milwaukee, WI, USA) and geranylgeraniol (85% pure, from Sigma Aldrich, Milwaukee, WI, USA).

### 3.2. Characterization of Diatomite

For the characteristics of diatomite, the following methods were used: X-ray diffractometry (XRD)—Empyrean X-ray diffractometer with Cu K$\alpha$ radiation source (Malvern Panalytical, Grovewood, UK); analysis of samples within the temperature range of 5–30° in 0.02° steps; mapping of elements—scanning electron microscopy (SEM) and EDX surface spectra SEM apparatus (JEOL Company, JSM-6010LA, Tokyo, Japan) with a secondary electron detector; elemental analysis performed with Epsilon3 energy dispersed X-ray fluorescence spectrometer (EDXRF) (Malvern Panalytical, Grovewood, UK); FT-IR infrared spectroscopy (Thermo Nicolet 380 apparatus, Malente, Germany); wavenumber range from 400 to 4000 cm$^{-1}$; specific surface area (SSA), total pore volume (TPV), and micropore

volume (MV)—nitrogen adsorption method at 350 °C using the QUADRASORB evoTM Gas Sorption Surface and Pore Size Analyzer (Quantachrome Instruments, Boynton Beach, FL, USA); prior to analyses, samples were degassed at 250 °C for 20 h in atm. $N_2$.

### 3.3. Method of Transformations of Geraniol and Analyses of Post-Reaction Mixtures

The syntheses were carried out in a glass reactor with a capacity of 25 cm³, which was equipped with a reflux condenser and a magnetic stirrer with a heating function. The ranges of the studied parameters were as follows: temperature range of 80–150 °C, catalyst content of 5–15 wt%, and reaction time from 15 min to 24 h. In order to perform qualitative and quantitative analyses, a sample of the post-reaction mixture was first centrifuged; then, it was dissolved in acetone at a ratio of 1:3.

Qualitative analyses were performed using the GC-MS method on a ThermoQuest apparatus with a Voyager detector and a DB-5 column (filled with phenylmethylsiloxanes, 30 m × 0.25 mm × 0.5 mm). The analysis parameters are as follows: helium flow at 1 mL/min; sample chamber temperature at 200 °C; detector temperature at 250 °C; oven temperature—isothermally for 2.5 min at 50 °C and then heating at a rate of 10 °C/min to 300 °C.

Quantitative analyses were performed with help of a Thermo Electron FOCUS chromatograph with an FID detector and TR-FAME column (cyanopropylphenyl packed, 30 m × 0.25 mm × 0.25 mm). The analysis parameters were as follows: helium flow at 0.7 mL/min; sample chamber temperature at 200 °C; detector temperature at 250 °C; oven temperature—isothermally for 7 min at 60 °C and then heating at a rate of 15 °C/min to 240 °C. The FID temperature was kept at a level of 250 °C.

The quantitative analyses of the products were performed using external and internal standard methods. In the case of the first method, 8-point calibration curves were performed for each compound within the concentration range of 0–33 wt%. After chromatographic analyses, the mass balances for each synthesis were prepared.

The subsequent process functions were used to describe individual syntheses:

(a).   Conversion of GA:

$$C_{GA} = \frac{number\ of\ moles\ of\ reacted\ GA}{number\ of\ moles\ of\ introduced\ GA} * 100\ [\%mol] \qquad (10)$$

(b).   Selectivity of BP:

$$S_{BP} = \frac{number\ of\ moles\ of\ product\ "x"}{number\ of\ moles\ of\ reacted\ substrate\ "s"} * 100\ [\%mol] \qquad (11)$$

The syntheses connected with geraniol transformations in the presence of the tested catalysts were carried out in a glass reactor with a capacity of 25 cm3, equipped with a reflux condenser and a magnetic stirrer with heating functions (Figure 14). The tested parameters were changed within the following ranges: temperature of 80–150 °C, catalyst content of 5–15 wt%, and reaction time from 15 min to 24 h.

### 3.4. Test Method

Method factors such as temperature, catalyst content, and reaction time were selected from our previous works. The tests control factors are illustrated in Table 3.

**Table 3.** Control parameters and values of the process.

| Control Parameters | Unit | Values | | |
|---|---|---|---|---|
| | | 1 | 2 | 3 |
| Temperature | (°C) | 80 | 110 | 150 |
| Catalyst concentration | (wt%) | 1 | 5 | 10 |
| Time | (h) | 0.25 | 12 | 24 |

Design of experiment (DOE) was used to decrease the test's number and to cut down inspection times. The tests were taken out in harmony with the full factorial layout. The response surface method (RSM) was used with a central composite model. The method provided a count of 27 experiments (Table 4). RSM is a combination of statistical and mathematical methods used for modeling. Additionally, it takes into regard a connection between the individual variable quantity of the process and the noted reactions.

**Table 4.** Factors of the designed experiment.

| Test nr. | Temp | Diatomite Concentration | Time | GA Conversion | BP Selectivity |
|---|---|---|---|---|---|
| - | (°C) | (wt%) | (h) | (% mol) | (% mol) |
| 1 | 80 | 1 | 0.25 | 22 | 16 |
| 2 | 80 | 1 | 12 | 28 | 24 |
| 3 | 80 | 1 | 24 | 35 | 27 |
| 4 | 80 | 5 | 0.25 | 27 | 25 |
| 5 | 80 | 5 | 12 | 31 | 38 |
| 6 | 80 | 5 | 24 | 36 | 41 |
| 7 | 80 | 10 | 0.25 | 33 | 30 |
| 8 | 80 | 10 | 12 | 38 | 36 |
| 9 | 80 | 10 | 24 | 41 | 42 |
| 10 | 110 | 1 | 0.25 | 38 | 36 |
| 11 | 110 | 1 | 12 | 43 | 47 |
| 12 | 110 | 1 | 24 | 64 | 58 |
| 13 | 110 | 5 | 0.25 | 68 | 55 |
| 14 | 110 | 5 | 12 | 73 | 67 |
| 15 | 110 | 5 | 24 | 88 | 76 |
| 16 | 110 | 10 | 0.25 | 89 | 64 |
| 17 | 110 | 10 | 12 | 90 | 77 |
| 18 | 110 | 10 | 24 | 99 | 89 |
| 19 | 150 | 1 | 0.25 | 30 | 43 |
| 20 | 150 | 1 | 12 | 40 | 45 |
| 21 | 150 | 1 | 24 | 60 | 56 |
| 22 | 150 | 5 | 0.25 | 65 | 51 |
| 23 | 150 | 5 | 12 | 70 | 63 |
| 24 | 150 | 5 | 24 | 80 | 72 |
| 25 | 150 | 10 | 0.25 | 88 | 62 |
| 26 | 150 | 10 | 12 | 98 | 77 |
| 27 | 150 | 10 | 24 | 99 | 87 |

Temp—temperature; GA—geraniol, BP—β-pinene.

The multinomial equation of second degree for determining the regression model value is as follows:

$$y = \beta_0 + \sum_{i=1}^{k} \beta_i x_i + \sum_{i=1}^{k} \beta_{ii} x_i^2 \pm \varepsilon \tag{12}$$

where

$y$    denotes the dependent variable (response);
$x_i$    indicates values of the i-th parameter;
$\beta_0, \beta_i, \beta_{ii}$    are the coefficients of regressions;
$\varepsilon$    denotes the acquired error.

For the computation process of model equations, Statistica software was used. The results of studies on the impact of process control factors (independent variables) on GA conversion and BP selectivity (dependent variables) are specified in Table 4. Columns 2 to 4 show control factors values (input data) for the test process. In addition, columns 5 to 7 show the answer values (output parameters).

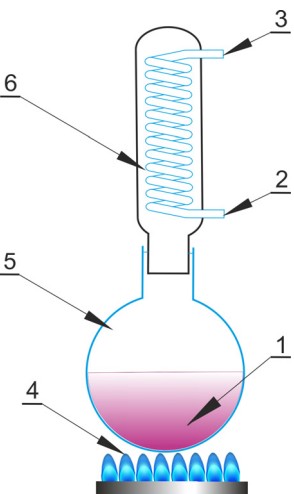

**Figure 14.** Scheme of the apparatus for carrying out the syntheses connected with the transformations of geraniol: 1—reaction mixture; 2—water outlet; 3—water inlet; 4—heating; 5—round bottom flask; 6—reflux condenser.

## 4. Conclusions

The use of statistical methods for yielding experiments and the optimization of the process permitted the following:

- A thorough examination of the assumed factor space with a minimum number of experiments performed;
- Full analysis of the errors of the model and experiments;
- Obtaining mathematical models of the process in analytical and graphical form.

By analyzing the obtained models for GA conversion, the optimum was achieved at 99 mol% for the following control parameters: temperature of 110 °C, catalyst concentration of 10.0 mol%, and time of 24 h. The optimum was reached for BP selectivity at 89 mol% for the following control parameters: 110 °C, catalyst concentration of 10.0 mol%, and time of almost 24 h. The optimum set of control parameters of the process for all output parameters includes the following: temperature of 80 °C, catalyst concentration of 1.0 wt%, and process time of 0.25 h.

Comparing the results obtained with those from preliminary studies [20], it was observed that the temperature value increased from 80 to 133 °C under optimal conditions. A similar trend was observed in the case of the catalyst concentration, where there was a tenfold increase in the amount of catalyst from 1 to 10 wt% of the process. However, the value of the selectivity of the transformation to BP decreased from 99 mol% to about 89 mol%. Nevertheless, using process optimization, the testing process can be simplified, reducing the time to obtain reliable results and lowering the cost of testing by reducing the number of tests required. Nevertheless, according to the described method, the testing process can be simplified, reducing the time to obtain reliable results again and lowering the cost of testing by reducing the number of tests required.

**Funding:** This research received no external funding.

**Data Availability Statement:** Not applicable.

**Conflicts of Interest:** The author declares no conflict of interest.

## Glossary

| | |
|---|---|
| RSM | Response surface methodology |
| BP | β-pinene |
| GA | Geraniol |
| RBF | Radial basis functions |
| DHA | Docosahexaenoic acid |
| SEM | Scanning electron microscopy |
| XRD | X-ray diffractometry |
| EDXRF | Energy dispersive X-ray fluorescence spectrometer |
| FT-IR | Infrared spectroscopy |
| SSA | Specific surface area |
| TPV | Total pore volume |
| MV | Micropore volume |
| GC-MS | Gas chromatography coupled to mass spectrometry |
| CGA | Conversion of geraniol |
| SBP | Selectivity of β-pinene |
| DOE | Design of experiment |
| VIF | Variance increase factor |
| CUR | Current |

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
