# Peer review of "Optimization of the Geraniol Transformation Process in the Presence of Natural Mineral Diatomite as a Catalyst"

_catalysts, doi:10.3390/catal13040777_

Round 1

Reviewer 1 Report

This manuscript does appear to be well discussed; however, more clarification on a few concepts are required. The following suggestions are useful.

1)     Instrumental results for SEM, XRD, FTIR, and Elemental Composition were found in the "Materials and Methods" section. Every characterization mentioned but there is no explanation.

2)    Authors should provide BP selectivity with catalyst concentration. Also, any possibility for other products formation?

3)    Authors explain more on how reaction time affects GA transformations and what temperature Does to GA transformations?

4)    The author should compare this work to previous research and highlight its technical advantages.

5) A full overhaul of the English language is required

Author Response

Answers for reviewer 1

Dear Reviewer#1,

Thank you very much for your in-depth review, correction and taking the time to comment on our paper under title:

“Optimization of the geraniol transformation process
in the presence of the natural mineral diatomite as a catalyst”

Manuscript ID: catalysts-2241920

My answers, point by point, explain the details of the revisions to the manuscript and our responses to reviewer comments.

Suggestions

Authors responses

Instrumental results for SEM, XRD, FTIR and Elemental Composition were found in the "Materials and Methods" section. Every characterization mentioned but there is no explanation.

Instrumental results for SEM, XRD, FTIR and elemental composition are included in the "2. Results and Discussions" section.

Authors should provide BP selectivity with catalyst concentration. Also, any possibility for other products formation?

Information on the selectivity of BP and the amount of catalyst is included in the applications.

Authors explain more on how reaction time affects GA transformations and what temperature Does to GA transformations?

Information on the possibility of the formation of other reaction products is included in the section "1. Introduction".

The author should compare this work to previous research and highlight its technical advantages.

The answer has been included in the proposals.

A full overhaul of the English language is required.

The English language has been corrected.

Reviewer 2 Report

The manuscript describes the optimization of the geraniol dehydration reaction catalyzed by diatomite, using the Response Surface Methodology. Although it is an interesting paper, some issues must be corrected before acceptance. 

1. The section "Results and discussion" seems to deal only with the optimization results. Direct experimental results are not given. The results of catalyst characterization are barely reported in the section "Materials and methods". No comparison is made with the results obtained with other catalysts.

2. A distinction must be made between Materials and Methods, Results, and Modeling. SEM micrographs and infrared spectra are results.

3. The reaction conditions are missing. Detailed information must be given concerning the reactor used, the reactant amounts, the sampling methodology, sampling preparation, method or methods used for the analysis of the reaction mixture samples, etc.

Some minor issues:

A glossary should be given.

The captions of figures and tables should contain the explanation of the acronyms used.

Page 3, line 122: the sentence "Physical (temperature and pH) and chemical (volume of substrate and enzyme)." makes no sense.

Figure 8: What Cur stands for? An explanation of the symbols and acronyms used should be given in the figure caption. I Assume that y = ###, in blue, is the value of the y-axis corresponding to the dashed line, in percentage. If it is so, how the author explains a conversion higher than 100 %?

Page 13, lines 261, 262: It seems to me that it must be the exact opposite.

Figure 9 must be divided into several figures and detailed explanations must be given in the corresponding captions, namely, the color codes used for elemental mapping. The XRD spectrum is missing.

Author Response

Answers for reviewer 2

Dear Reviewer#2,

Thank you very much for your in-depth review, correction and taking the time to comment on our paper under title:

“Optimization of the geraniol transformation process
in the presence of the natural mineral diatomite as a catalyst”

Manuscript ID: catalysts-2241920

My answers, point by point, explain the details of the revisions to the manuscript and our responses to reviewer comments.

Suggestions

Authors responses

The section "Results and discussion" seems to deal only with the optimization results. Direct experimental results are not given. The results of catalyst characterization are barely reported in the section "Materials and methods". No comparison is made with the results obtained with other catalysts.

The results on catalyst characterization have been completed and placed in Chapter 2 Results and Discussion.  The second chapter deals only with the optimization results, since the results from the preliminary studies were placed in the author's article:

Fajdek-Bieda A, Wróblewska A, MiÄ…dlicki P, Konstanciak A. Conversion of Geraniol into Useful Value-Added Products in the Presence of Catalysts of Natural Origin: Diatomite and Alum. Materials (Basel). 2022 Mar 26;15(7):2449. doi: 10.3390/ma15072449. PMID: 35407782; PMCID: PMC9000025.

 No comparison was made with results obtained with other catalysts, as this was not the subject of this work.

A distinction must be made between Materials and Methods, Results, and Modeling. SEM micrographs and infrared spectra are results.

The comment has been corrected.

The reaction conditions are missing. Detailed information must be given concerning the reactor used, the reactant amounts, the sampling methodology, sampling preparation, method or methods used for the analysis of the reaction mixture samples, etc.

The information is supplemented in "3. Materials and Method".

A glossary should be given.

A glossary of abbreviations is provided at the end of the publication.

The captions of figures and tables should contain the explanation of the acronyms used.

Figure and table captions have been supplemented with an explanation of the abbreviations used.

Page 3, line 122: the sentence "Physical (temperature and pH) and chemical (volume of substrate and enzyme)." makes no sense.

The sentence has been deleted.

Figure 8: What Cur stands for? An explanation of the symbols and acronyms used should be given in the figure caption. I Assume that y = ###, in blue, is the value of the y-axis corresponding to the dashed line, in percentage. If it is so, how the author explains a conversion higher than 100 %?

After reanalyzing the data, the above blunder has been corrected. The abbreviation Cur is explained in the glossary of abbreviations.

Page 13, lines 261, 262: It seems to me that it must be the exact opposite.

The sentence was corrected in the article.

Figure 9 must be divided into several figures and detailed explanations must be given in the corresponding captions, namely, the color codes used for elemental mapping. The XRD spectrum is missing.

The Reviewer's suggestions have been incorporated into the paper.
